# Private GANs, Revisited

**Alex Bie**[*]
University of Waterloo
yabie@uwaterloo.ca

**Gautam Kamath**
University of Waterloo
g@csail.mit.edu

**Guojun Zhang**
Huawei Noah's Ark Lab
guojun.zhang@huawei.com

## Abstract

We show that with improved training, the standard approach for differentially private GANs – updating the discriminator with noisy gradients – achieves or competes with state-of-the-art results for private image synthesis. Existing instantiations of this approach neglect to consider how adding noise only to discriminator updates disrupts the careful balance between generator and discriminator necessary for successful GAN training. We show that a simple fix restores parity: taking more discriminator steps between generator steps. Furthermore, with the goal of restoring parity, we experiment with further modifications to improve discriminator training and see further improvements in generation quality. For MNIST at a privacy budget of $\varepsilon = 10$, our private GANs improve the record FID from 48.4 to 13.0, as well as downstream classifier accuracy from 83.2% to 95.0%.

## 1 Introduction

A recent line of work has studied leveraging deep generative models to produce differentially private synthetic data. Initial efforts focused on privatizing generative adversarial networks (GANs) [14] by using differentially private stochastic gradient descent (DPSGD) [1] to update the GAN discriminator – an approach referred to as *DPGAN* [33, 6, 27]. Follow-up work has studied alternative approaches to privatizing GANs [18, 20, 8, 31], as well as other generative modelling frameworks, such as maximum mean discrepancy [15, 29], Sinkhorn divergences [7], and energy-based models [9].

Towards generating high-dimensional, complex data, these studies have primarily focused on image synthesis, which has served as the testbed of choice for research in (non-private) generative modelling. For the task of labelled image synthesis, the literature has corroborated that departing from the baseline DPGAN approach, either in the privatization scheme, or modelling framework altogether, leads to significant improvements in generation quality. Proposed explanations attribute these results to inherent limitations of the DPGAN framework, suggesting that: (1) privatizing discriminator training is sufficient for privacy, but may be overkill when only the generator needs to be released [20]; or (2) adversarial objectives may be unsuited for training under privacy [7].

We demonstrate that the reported poor results of DPGANs should not be attributed to inherent limitations of the framework, but rather, training issues. More precisely: we propose that the *asymmetric noise addition* in DPGANs (adding noise to discriminator updates only) weakens the discriminator relative to the generator, disrupting the careful balance necessary for successful GAN training. To account for this, we propose to take more discriminator steps between generator updates. With this change, DPGANs improve significantly (see Figure 1), achieving or competing with state-of-the-art results in private image synthesis.

Furthermore, we demonstrate that this perspective on private GAN training ("restoring parity to a discriminator weakened by DP noise") is an effective heuristic for improving private GANs. We make other modifications to discriminator training – large batch sizes and discriminator step schedules – and see futher improvements.

---

[*]Work performed in part while interning at Huawei.

NeurIPS 2022 Workshop on Synthetic Data for Empowering ML Research.

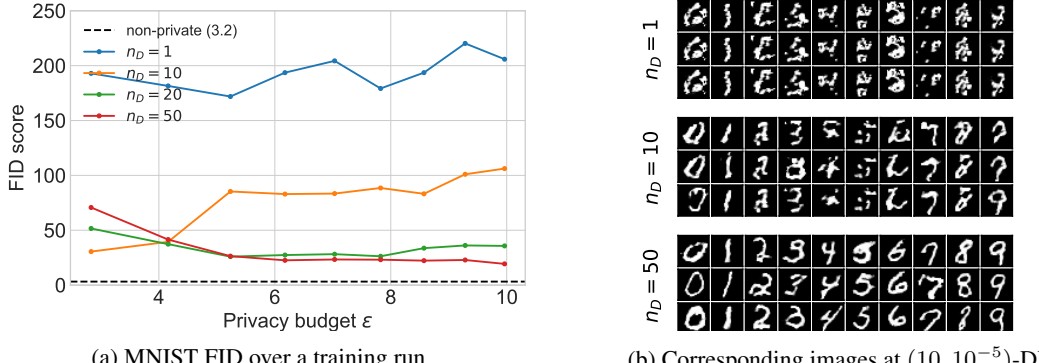

(a) MNIST FID over a training run  (b) Corresponding images at $(10, 10^{-5})$-DP

Figure 1: Our DPGAN results on labelled MNIST synthesis at $(10, 10^{-5})$-DP. **(a)** FID over training runs varying $n_{\mathcal{D}}$. We find that increasing $n_{\mathcal{D}}$, the number of discriminator steps taken between generator steps, significantly improves image synthesis results. Using $n_{\mathcal{D}} = 50$ instead of $n_{\mathcal{D}} = 1$ on our baseline DPGAN improves FID from $205.9 \rightarrow 19.4$, which is also an improvement over the record FID of $48.4$ from [7]. $n_{\mathcal{D}} = 50$ also improves downstream classification accuracy to $92.9\%$ (see Figure 2a), improving over the the record accuracy of $83.2\%$ from [7]. **(b)** Corresponding synthesized images. We observe that large $n_{\mathcal{D}}$ improves visual quality, and low $n_{\mathcal{D}}$ leads to mode collapse.

## 2  Preliminaries

Our goal is to train a generative model on sensitive data that is safe to release, i.e., it does not leak the secrets of individuals in the training dataset. We do this by ensuring the training algorithm $\mathcal{A}$ – which takes as input the sensitive dataset $D \in \mathcal{U}$ and returns the parameters of a trained (generative) model $\theta \in \Theta$ – satisfies differential privacy.

**Definition 1** (Differential Privacy [13])**.** A randomized algorithm $\mathcal{A} : \mathcal{U} \to \Theta$ is $(\varepsilon, \delta)$-*differentially private* if for every pair of neighbouring datasets $D, D' \in \mathcal{U}$, we have

$$\mathbb{P}\{\mathcal{A}(D) \in S\} \leq \exp(\varepsilon) \cdot \mathbb{P}\{\mathcal{A}(D') \in S\} + \delta \qquad \text{for all } S \subseteq \Theta.$$

We adopt the add/remove definition of DP, and say two datasets $D$ and $D'$ are neighbouring if they differ in at most one entry, that is, $D = D' \cup \{x\}$ or $D' = D \cup \{x\}$.

We highlight one convenient property of DP, known as *closure under post-processing*, which says that interacting with a privatized model (e.g., using it to compute gradients on non-sensitive data, drawing samples from it) does not lead to any further privacy violation.

**DPSGD.**  A gradient-based training algorithm can be privatized by employing *differentially private stochastic gradient descent (DPSGD)* [25, 5, 1] as a drop-in replacement for SGD. DPSGD involves clipping per-example gradients and adding Gaussian noise to their sum, which effectively bounds and masks the contribution of any individual point to the final model parameters. Privacy analysis of DPSGD follows from a number of classic tools in the DP toolbox: the Gaussian mechanism, privacy amplification by subsampling, and composition [12, 1, 4, 32]. We employ the DPSGD analysis of [21] implemented in the Opacus library [34].

**DPGANs.**  Algorithm 1 details the training algorithm for DPGANs, which is effectively an instantiation of DPSGD. Note that only gradients for the discriminator $\mathcal{D}$ must be privatized (via clipping and noise), and not those for the generator $\mathcal{G}$ due to post-processing.

## 3  Frequent discriminator steps improves private GANs

In this section, we discuss our main finding: the number of discriminator steps taken between each generator step ($n_{\mathcal{D}}$ from Algorithm 1) plays a significant role in the success of private GAN training. For a fixed setting of DPSGD hyperparameters, there is an optimal range of values for $n_{\mathcal{D}}$ that

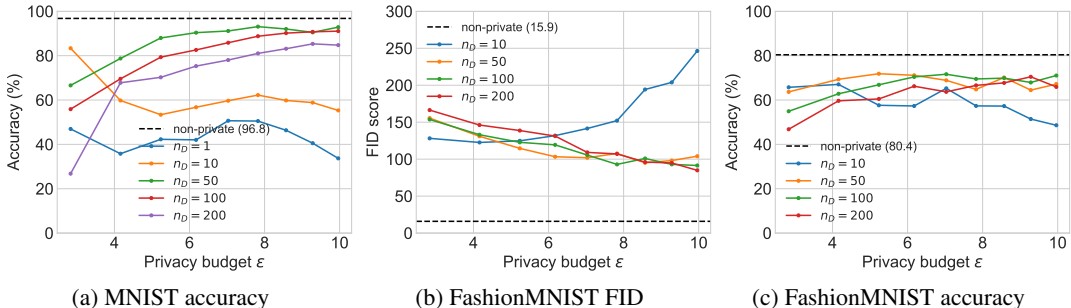

(a) MNIST accuracy      (b) FashionMNIST FID      (c) FashionMNIST accuracy

Figure 2: DPGAN results targeting $(10, 10^{-5})$-DP under different discriminator update frequencies $n_{\mathcal{D}}$. **(a)** MNIST downstream classification accuracy over training runs, which mirrors FID scores (seen in Figure 1a). Going from to $n_{\mathcal{D}} = 1$ to $n_{\mathcal{D}} = 50$ improves accuracy from $33.7\% \rightarrow 92.9\%$. **(b) & (c)** We obtain similar results for FashionMNIST training runs. Note that the optimal $n_{\mathcal{D}}$ is higher (around $n_{\mathcal{D}} \approx 100$). At $n_{\mathcal{D}} = 100$, we obtain an FID of 91.5 and accuracy of 71.1%, which compares favourably to the record FID of 128.3 and record accuracy of 75.5% reported in [7] for $(10, 10^{-5})$-DP generation of FashionMNIST.



$t = 50\text{K}$      $t = 100\text{K}$      $t = 150\text{K}$      $t = 200\text{K}$

Figure 3: Evolution of samples drawn during a training run at $n_{\mathcal{D}} = 10$, when targeting $(10, 10^{-5})$-DP. This setting reports its best FID and downstream accuracy at $t = 50\text{K}$ iterations ($\varepsilon \approx 2.85$). As training progresses, we observe mode collapse alongside deterioration in evaluation metrics.

maximizes generation quality, in terms of both visual quality and utility for downstream classifier training. This value is often quite large ($n_{\mathcal{D}} \approx 100$ in some cases).

### 3.1 Experimental details

**Setup.** We focus on labelled generation of MNIST and FashionMNIST. To build a strong baseline, we begin from an open source PyTorch [23] implementation[2] of DCGAN [24] that performs well non-privately, and copy their training recipe. We adapt their architecture to our purposes: removing BatchNorm layers (which not compatible with DPSGD) and adding label embedding layers to enable labelled generation. Training this configuration non-privately yields labelled generation that achieves FID scores of 3.2 on MNIST and 15.9 on FashionMNIST. Finally, we note that these models are not small: $\mathcal{D}$ and $\mathcal{G}$ have 1.72M and 2.27M trainable parameters respectively.

**Privacy implementation.** To privatize training, we use Opacus [34] which implements per-example gradient computation and the RDP accounting of [21]. For our baseline setting, we use the following DPSGD hyperparameters: we keep the non-private (expected) batch size $B = 128$, and use a noise scale $\sigma = 1$ and clipping norm $C = 1$. Under these settings, we have the budget for $T = 450,000$ discriminator steps when targeting $(10, 10^{-5})$-DP.

**Evaluation.** We evaluate our generative models by examining the *visual quality* and *utility for downstream tasks* of generated images. Following prior work, we measure visual quality by computing the Fréchet Inception Distance (FID) [16] between 60,000 generated images and 10,000 real images from the test set.[3] To measure downstream task utility, we again follow prior work, and train a CNN classifier on 60,000 generated image-label pairs and report its accuracy on the real test set.

---

[2]Courtesy of Hyeonwoo Kang (`https://github.com/znxlwm`). Code available at this link.

[3]We use an open source PyTorch implementation to compute FID: `https://github.com/mseitzer/pytorch-fid`

## 3.2 More frequent discriminator steps improves generation quality

We plot in Figures 1a and 2 the evolution of FID and downstream accuracy during DPGAN training for both MNIST and FashionMNIST, under varying discriminator update frequencies $n_\mathcal{D}$. The effect of this parameter has outsized impact on the final results. For MNIST, $n_\mathcal{D} = 50$ yields the best results; on FashionMNIST, the best FID is obtained at $n_\mathcal{D} = 200$ and the best accuracy at $n_\mathcal{D} = 100$.

We emphasize that increasing $n_\mathcal{D}$, the *frequency* of discriminator steps relative to generator steps, does not affect the privacy cost $\varepsilon$ of Algorithm 1. For any setting of $n_\mathcal{D}$, we perform the same number of discriminator steps (noisy gradient queries on real data) – what changes is the total number of generator steps taken over the course of training, which is reduced by a factor of $1/n_\mathcal{D}$.

## 3.3 Private GANs are on a path to mode collapse

In Figures 1a and 2a, we observe that at low discriminator update frequencies ($n_\mathcal{D} = 10$), the best FID and accuracy scores occur early in training, well before the privacy budget we are targeting is exhausted.[4] At $50,000$ discriminator steps ($\varepsilon \approx 2.85$), $n_\mathcal{D} = 10$ has better FID (30.6) and accuracy (83.3%) than the other settings. However, these results degrade by the end of training. In Figure 3, we plot the evolution of generated images in this setting over the course of training, and observe qualitative evidence of mode collapse, co-occurring with deterioration in evaluation metrics.

These results suggest that *fixing other DPSGD hyperparameters, there is an optimal setting for the discriminator step frequency $n_\mathcal{D}$ that strikes a balance between:* (1) being too low, causing the model performance peak early in training and undergo mode collapse; resulting in all subsequent training to consume additional privacy budget *without improving the model*; and (2) being too high, preventing the generator from taking enough steps to converge before the privacy budget is exhausted (an example of which is pictured in Figure 2a). Striking this balance results in the most effective utilization of privacy budget towards improving the generator.

## 4 Why does taking more steps help?

Figure 4 compares the accuracy of the GAN discriminator (on held-out real and fake examples) immediately before each generator step, between non-private training and private training for different settings of $n_\mathcal{D}$. Non-privately, discriminator accuracy stays around 60% throughout training. Introducing DP ($n_\mathcal{D} = 1$) leads to a qualitative difference: DP causes discriminator accuracy to drop to 50% immediately, and never recovers. For other settings of $n_\mathcal{D}$, we make three observations: (1) Larger $n_\mathcal{D}$ corresponds to higher accuracy; (2) The generator improves during the periods in which the discriminator stays above 50% accuracy; and (3) accuracy decreases throughout training, and the degradation of results co-occurs with accuracy of the discriminator dropping below 50%.

Based on these observations, we propose the following explanation on why more steps helps:

- Generator improvement occurs when our discriminator is capable of distinguishing between real and fake data.

- The asymmetric noise addition introduced by DP to the discriminator makes the task difficult.

- Allowing the discriminator to train longer on a fixed generator improves its accuracy, recovering the non-private case where the generator and discriminator are balanced.

## 5 Better generators via better discriminators

Our proposed explanation in Section 4 provides a suggestion for improving GAN training: effectively use our privacy budget to maximize the number of generator steps taken when the discriminator has a high accuracy. We experiment with modifications to the private GAN training recipe to improve discriminator accuracy, which translate to improved generation.

---

[4]This observation has been reported in [22], serving as motivation for their remedy of taking a mixture of intermediate models encountered in training. We are not aware of any mentions of this aspect of DPGAN training in papers reporting DPGAN baselines for labelled image synthesis.

## 5.1 Larger batch sizes

Several recent works have demonstrated that for classification tasks, large batch sizes improve the performance of DPSGD, after tuning for the optimal noise scale $\sigma$ accordingly [28, 2, 11]. GAN training is typically conducted with small batch sizes (for example, DCGAN uses $B = 128$, which we adopt; StyleGAN uses $B = 32$). Therefore it is interesting to see if large batch sizes indeed improve private GAN training. We scale up batch sizes, considering $B \in \{64, 128, 512, 2048\}$, and search for the optimal noise scale $\sigma$ and $n_{\mathcal{D}}$. We target two settings, the high privacy ($\varepsilon = 1$) and low privacy ($\varepsilon = 10$) regimes.

We report the best results from our hyperparameter search in in Table 1. We find that larger batch sizes leads to improvements: for $\varepsilon = 10$, the best MNIST and FashionMNIST results are achieved at $B = 2048$. For $\varepsilon = 1$, the best results are achieved at $B = 512$.

## 5.2 Discriminator step scheduling

Our observations from Section 3 and 4 motivate us to consider a discriminator step frequency schedule. As pictured in Figure 4, discriminator accuracy drops during training as the quality of generated images improve, which co-occurs with degradation of results. In this scenario, we want to take more steps to improve the discriminator. However, using a large discriminator update frequency at the beginning of training is wasteful – as evidenced by the fact that low $n_{\mathcal{D}}$ achieves the best FID and accuracy early in training. Hence we propose to start at a low discriminator update frequency, and ramp up when our discriminator is performing poorly.

The accuracy on real data must be released with DP. While this is feasible, it introduces the additional problem of having to find the right split of privacy budget for the best performance. We observe that discriminator accuracy is related to discriminator accuracy on fake samples only (which are free to evaluate on, by post-processing). Hence we use it as a proxy to assess discriminator performance.

The step schedule is parameterized by two terms, $\beta$ and $d$. $\beta$ is the decay parameter used to compute the exponential moving average (EMA) of discriminator accuracy on fake batches before each generator update. We use $\beta = 0.99$ in all settings. $d$ is the accuracy threshold that upon falling below, we increase the update frequency. We try $d = 0.6$ and $d = 0.7$, finding that $0.7$ works better for large batches. Additionally, we promise a grace period of $2/(1 - \beta) = 200$ generator steps before moving on to the next update frequency. This formula is motivated by the fact that $\beta$-EMA's value is primarily determined by its last $2/(1 - \beta)$ observations.

The additional benefit of the step schedule is that it means we do not have to search for the optimal update frequency. Although the step schedule introduces the extra hyperparameter of the threshold $d$, we found that these two settings ($d = 0.6$ and $0.7$) were sufficient to improve over results of a much more extensive hyperparameter search.

## 5.3 Comparison with state of the art

Table 1 summarizes our best experimental settings. In the low privacy/high $\varepsilon$ regime, most of our results are dramatically better than prior work[4] – for example, decreasing FID from 48.4 to 13.0 and increasing accuracy from 83.2% to 95.0% on MNIST. In the high privacy/low $\varepsilon$ regime, improvements are not quite as extreme, but can still be significant (FID for MNIST), and only compare negatively to state-of-the-art for accuracy on Fashion MNIST. Broadly speaking, we see an improvement from taking larger batch sizes, and then another with an adaptive step schedule. We provide some example generated images in Figures 5 and 6 for $\varepsilon = 10$, and Figures 7 and 8 for $\varepsilon = 1$.

## 6 Discussion and related work

**DP generative modelling.** Differentially private GANs (DPGANs) were introduced by Xie et al. [33]. Subsequent works proposed alternative privatizations schemes for GANs settings [27, 18, 20, 8, 31]; they did not consider a significantly larger number of discriminator steps per generator step.

---

[4]We do not compare with two recent works on private generative models [9, 17], as we believe there are gaps in their privacy analyses. This has been confirmed by the authors of [17], and the sketch of an argument regarding non-privacy of [9] has been shared with us by others [3].

| Privacy Level | Method | Reported In | MNIST | | FashionMNIST | |
|---|---|---|---|---|---|---|
| | | | FID | Acc.(%) | FID | Acc.(%) |
| $\varepsilon = \infty$ | Real data | (This work) | 1.0 | 99.2 | 1.5 | 92.5 |
| | GAN | | 3.2 | 96.8 | 15.9 | 80.4 |
| | DPGAN[5] | [8] | 179.16 | 63 | 243.80 | 50 |
| | | [20] | 304.86 | 80.11 | 433.38 | 60.98 |
| $\varepsilon = 10$ | GS-WGAN | [8] | 61.34 | 80 | 131.34 | 65 |
| | PATE-GAN | [20] | 253.55 | 66.67 | 229.25 | 62.18 |
| | G-PATE | [20] | 150.62 | 80.92 | 171.90 | 69.34 |
| | Datalens | [31] | 173.50 | 80.66 | 167.68 | 70.61 |
| | DP-MERF | [7] | 116.3 | 82.1 | 132.6 | **75.5** |
| | DP-Sinkhorn | [7] | 48.4 | 83.2 | 128.3 | 75.1 |
| | DPGAN | | 19.4 | 92.9 | 91.5 | 71.1 |
| | + large batches | (This work) | 13.2 | 94.3 | 66.7 | 72.1 |
| | + step schedule | | **13.0** | **95.0** | **56.8** | 74.8 |
| | DPGAN | [20] | 470.20 | 40.36 | 472.03 | 10.53 |
| $\varepsilon = 1$ | GS-WGAN | [20] | 489.75 | 14.32 | 587.31 | 16.61 |
| | PATE-GAN | [20] | 231.54 | 41.68 | 253.19 | 42.22 |
| | G-PATE | [20] | 153.38 | 58.80 | 214.78 | 58.12 |
| | Datalens | [31] | 186.06 | 71.23 | 194.98 | 64.78 |
| | DP-MERF | [17] | 118.3 | 80.5 | **102.1** | **74.6** |
| | DP-MERF | [29][6] | - | 80.7 | - | 73.9 |
| | DP-HP | [29][6] | - | **81.5** | - | 72.3 |
| | DPGAN | | 91.7 | 77.4 | 151.9 | 65.0 |
| | + large batches | (This work) | 66.1 | 73.7 | 153.2 | 66.6 |
| | + step schedule | | **56.2** | 80.1 | 121.8 | 68.0 |

Table 1: We gather previously reported results in the literature on the performance of various methods for labelled generation of MNIST and FashionMNIST. For downstream accuracy, we report the best accuracy among classifiers they use, and compare against our CNN classifier accuracy.

Other private generative models include VAEs [10], maximum mean discrepancy [15], Sinkhorn divergences [7], energy-based models [9], and normalizing flows [30]. We show that a well-tuned DPGAN competes with or outperforms these approaches.

**Custom approaches versus a well-tuned DPSGD.** An ongoing debate pertains to the best techniques and architectures for private ML. Roughly speaking, there are two schools of thought. One investigates novel architectures for privacy, which may be outperformed by more traditional approaches in the non-private setting. Some examples include [7, 15, 29], a variety of generative models specifically designed to be compatible with differential privacy. The other focuses on searching within the space of tried-and-tested methods that are understood to work well non-privately. Some examples include the works of [11, 19, 35], who demonstrate that, similar to the non-private setting, large-scale CNN and Transformer architectures can achieve state-of-the-art results for image classification and NLP tasks. The primary modifications to the pipeline are along the lines of changing the batch size, modifying normalization layers, etc., most of which would be explored in a proper hyperparameter search in the non-private setting. Our work fits into the latter line: we show that novel generative models introduced for privacy can be outperformed by GANs trained with well-tuned DPSGD.

**Tabular data.** Our investigation focused on image datasets, namely MNIST and Fashion MNIST, while many important applications of private data generation involve *tabular data*. While [26] find that private GAN-based approaches fail to preserve even basic statistics in these settings, we believe that our techniques may yield similar improvements.

---

[5]We group per-class unconditional GANs together with conditional GANs under the DPGAN umbrella.

[6]These results are presented graphically in the paper. Exact numbers can be found in their code.

# 7 Conclusion

Our most important contribution is to show that private GANs have been underrated by the research community, and in fact can achieve state-of-the-art results by properly exploring the search space. We hope and anticipate this will inspire the community to revisit private GANs, and quickly improve upon our results.

## Acknowledgments and Disclosure of Funding

Work performed in part while AB was interning at Huawei. AB and GK are supported by an NSERC Discovery Grant, a University of Waterloo Startup Grant, an unrestricted gift from Google, and an unrestricted gift from Apple. AB is supported by a Vector Scholarship in Artificial Intelligence.

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

# A Extra figures

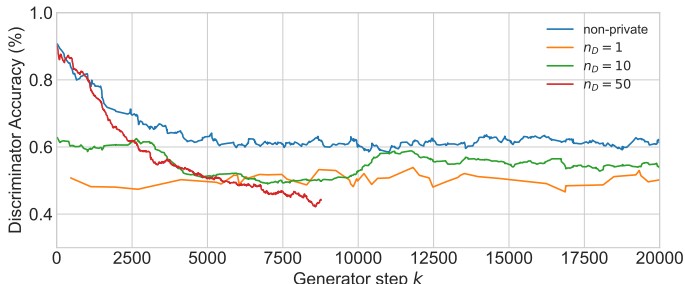

Figure 4: Discriminator accuracy before each generator step. While non-privately the discriminator maintains 60% accuracy, the private discriminator with $n_{\mathcal{D}} = 1$ is effectively a random guess. Increasing the number of discriminator steps recovers the discriminator's advantage early on.

---

**Algorithm 1** TrainDPGAN$(D; \cdot)$

---

1: **Input:** Labelled dataset $D = \{(x_j, y_j)\}_{j=1}^n$. Discriminator $\mathcal{D}$ and generator $\mathcal{G}$ initializations $\phi_0$ and $\theta_0$. Optimizers OptD, OptG. Privacy parameter $\delta$. Hyperparameters: $n_{\mathcal{D}}$ ($\mathcal{D}$ steps per $\mathcal{G}$ step), $T$ (# of $\mathcal{D}$ steps), $B$ (expected batch size), $C$ (clipping norm), $\sigma$ (noise multiplier).
2: $q \leftarrow B/|D|$ and $t, k \leftarrow 0$      $\triangleright$ calculate sampling rate $q$, initialize counters
3: **while** $t < T$ **do**      $\triangleright$ update $\mathcal{D}$ with DPSGD
4:      $S_t \sim \text{PoissonSample}(D, q)$    $\triangleright$ sample a real batch $S_t$ by including each $(x, y) \in D$ w.p. $q$
5:      $\widetilde{S}_t \sim \mathcal{G}(\cdot; \theta_k)^B$      $\triangleright$ sample fake batch $\widetilde{S}_t$
6:      $g_{\phi_t} \leftarrow \sum_{(x,y) \in S_t} \text{clip}_C \left( \nabla_{\phi_t}(-\log(\mathcal{D}(x, y; \phi_t))) \right)$
         $+ \sum_{(\widetilde{x}, \widetilde{y}) \in \widetilde{S}_t} \text{clip}_C \left( \nabla_{\phi_t}(-\log(1 - \mathcal{D}(\widetilde{x}, \widetilde{y}; \phi_t))) \right)$    $\triangleright$ clipped per-example gradients
7:      $\widehat{g}_{\phi_t} \leftarrow \frac{1}{2B}(g_{\phi_t} + z_t)$, where $z_t \sim \mathcal{N}(0, C^2\sigma^2 I))$      $\triangleright$ add Gaussian noise
8:      $\phi_{t+1} \leftarrow \text{OptD}(\phi_t, \widehat{g}_{\theta_t})$ and $t \leftarrow t + 1$
9:      **if** $n_{\mathcal{D}}$ divides $t$ **then**      $\triangleright$ perform $\mathcal{G}$ update every $n_{\mathcal{D}}$ steps
10:          $\widetilde{S}'_t \sim \mathcal{G}(\cdot; \theta_k)^B$
11:          $g_{\theta_k} \leftarrow \frac{1}{B} \sum_{(\widetilde{x}, \widetilde{y}) \in \widetilde{S}'_t} \nabla_{\theta_k}(-\log(\mathcal{D}(\widetilde{x}, \widetilde{y}; \phi_t)))$
12:          $\theta_{k+1} \leftarrow \text{OptG}(\theta_k, g_{\theta_k})$ and $k \leftarrow k + 1$
13:      **end if**
14: **end while**
15: $\varepsilon \leftarrow \text{PrivacyAccountant}(T, \sigma, q, \delta)$      $\triangleright$ compute privacy budget spent
16: **Output:** Final $\mathcal{G}$ parameters $\theta_k$. $(\varepsilon, \delta)$-DP guarantee.

---

# B Generated samples

We provide a few non-cherrypicked samples from MNIST and FashionMNIST at $\varepsilon = 1$ and $\varepsilon = 10$.

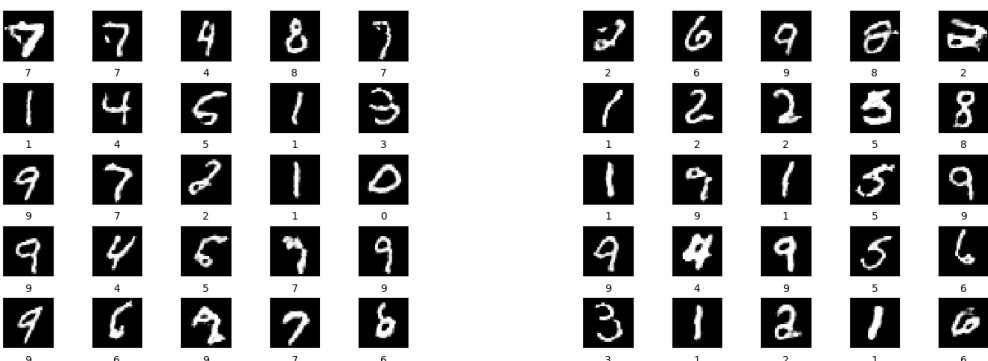

Figure 5: Some non-cherrypicked MNIST samples from our method, $\varepsilon = 10$.

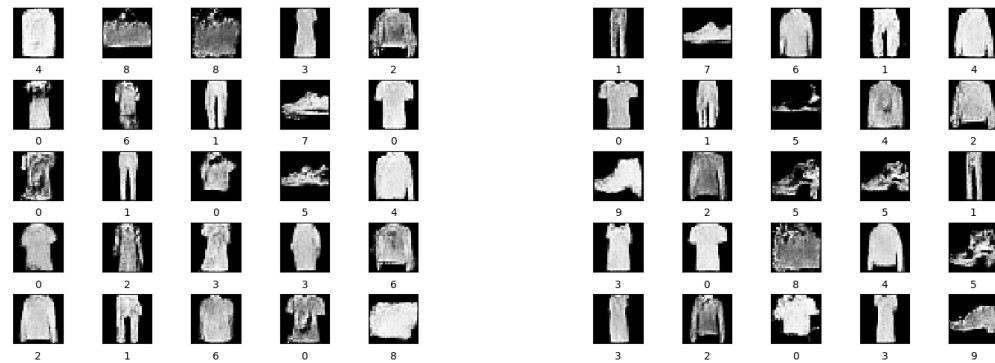

Figure 6: Some non-cherrypicked FashionMNIST samples from our method, $\varepsilon = 10$.

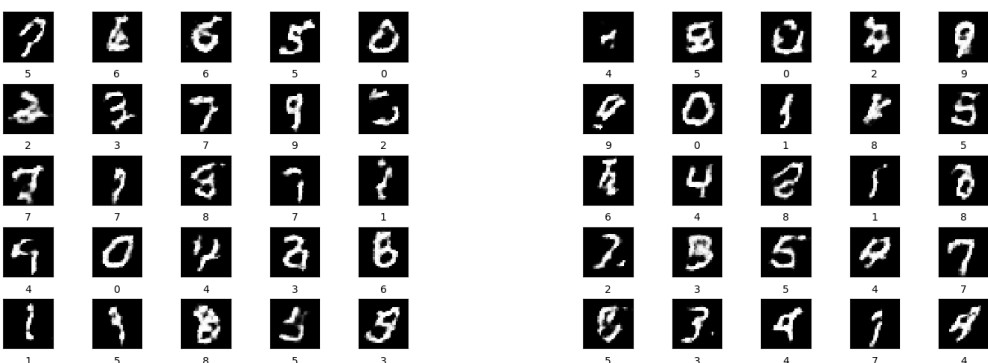

Figure 7: Some non-cherrypicked MNIST samples from our method, $\varepsilon = 1$.

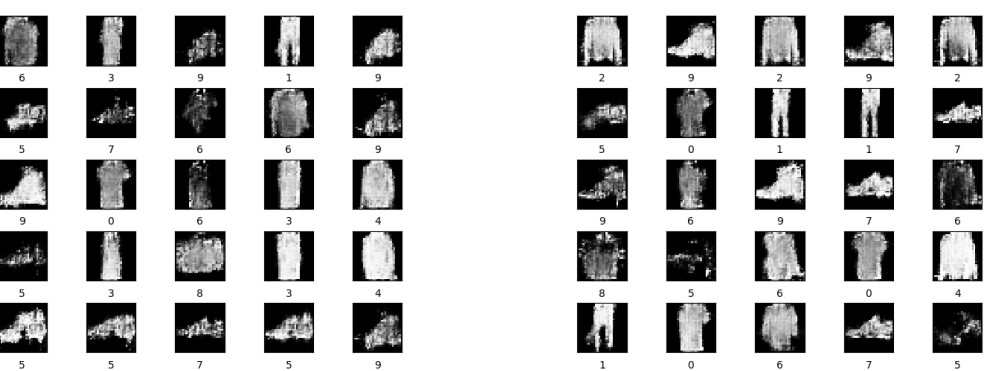

Figure 8: Some non-cherrypicked FashionMNIST samples from our method, $\varepsilon = 1$.

