# OpenReview forum: "Private GANs, Revisited"
_NeurIPS.cc/2022/Workshop/SyntheticData4ML — Neurips 2022 SyntheticData4ML_

### Official Review · Reviewer_VMyu · 2022-10-15
**An improved DPGAN training with a significant improvement**

**Rating:** 8
**Confidence:** 3

**Review:**

**Summary**: This paper proposes a simple method to improve DPGAN training. The author analyzes DPGAN from a new perspective, and claims that the balance between the generator and discriminator is destroyed after adding noise to the discriminator, resulting in the observed inferior results of DPGAN in previous work. To address the problem, the author proposes to restore parity by simply increasing the number of discriminator steps between generator steps. Based on the empirical insight that the extra steps induced ($n_{D}$) enable the discriminator to recover its accuracy to the non-private case, which could also benefit the generator later, the author further propose larger batch training and discriminator step scheduling to effectively exploit the privacy budget. The experimental results demonstrate a significant improvement over the SOTA on MNIST.

**Pros**:
* The proposed method is quite simple. The underlying insights (i.e., Section 4) also make sense.
* The experimental results demonstrate a significant improvement over the SOTA.

**Cons**:
* I suggest the author polish their writing in the next version. Please give more details or explanations about notations and symbols **when they first appear**, so that readers from other areas could understand the content. For example, In line 11, what does $\epsilon$ represent, privacy budget? etc.
* In lines 89-91, the author mentions that varying the frequency of discriminator steps relative to the generator steps does not affect the privacy cost. What does the privacy cost mean here? How about the time complexity (or computational cost) of the proposed method, since you increase the number of discriminator steps (from 1 -> 50/100)? What's the result (e.g., FID) of the baseline and proposed method within the same running time? Please discuss more here.
* In Figure 2 (b) & (c), please add the result of $n_{D}=1$.
* As Table 1 shown, it seems that the proposed method does not work well on FashionMNIST, while *very* well on MNIST. It may raise a concern about the performance on some complicated datasets (e.g., I noticed that [1] also conducted experiments on CelebA). It would be better to validate the proposed method on it if the author plans to submit it to the top conference.
* In Section 4, the authors claim that multiple $n_{D}$ enables the private discriminator (accuracy) to recover the balance between G and D in the non-private case. It makes sense to me. I suggest the authors plot the discriminator accuracies (you should have 50 accuracies if $n_{D}=50$) between two consecutive G's updates. Why the red line stops at around 9K steps in Figure 4? It's weird. I don't think you should constrain the privacy budget in this figure. Moreover, Section 4 could serve as the (empirical) motivation of the proposed method. Is there any theoretical insights behind it? If not, I suggest the author conduct comprehensive empirical illustrations/experiments to convince either readers or reviewers in this part.

**Overall**, the idea is simple with a significant improvement, which means it may touch the essence of the studied problem. A better writing, analysis (e.g., Section 4), and further experiments on FashionMNIST and CelebA could be considered. If addressed well, I believe this work could have a high impact in this area. BWT, the author could only take useful advice since I do not have a background in DP. Based on the current draft, I recommend a clear acceptance.

[1] Don’t Generate Me: Training Differentially Private Generative Models with Sinkhorn Divergence.

---

### Official Review · Reviewer_i3CV · 2022-10-16

**Rating:** 6
**Confidence:** 4

**Review:**

The paper addresses the problem of training differentially private (DP) GANs. The paper illustrates an interesting phenomenon: by simply increasing the number of discriminator steps between generator steps, we can get a better fidelity-privacy tradeoff. In addition, the paper proposes a clever approach to automatically adjust the discriminator step across training without incurring additional privacy costs. Experiments show that the proposed approach (plus larger batch sizes) can significantly improve the privacy-fidelity tradeoff of DP GANs.

Overall, the ideas are interesting and useful, and technically sound. I have a few suggestions for improving the paper:

* The figures can be better organized. Figures 1 (a) and 2 are from the same set of experiments. It is confusing to put them in two different figures, especially when readers are reading the experimental section.

* In Figure 1, it is noted that the results are over a training run. But in Figure 2, there is no such explanation. It is unclear whether each point or line corresponds to one run. (Although I believe it's the former).

* It might be better to merge Section 4 into 3 as they are both explaining the 1st proposed approach.

---

### Official Review · Reviewer_Qn9s · 2022-10-18
**Well-written paper with some impact**

**Rating:** 7
**Confidence:** 2

**Review:**

This is a well-written paper with clear objectives and contributions. The paper describes improvements to private GANs with a simple but effective change to the training procedure.  The impact is that the new results with private results outperform state-of-the-art alternatives.

The improvement associated with increasing the number of discriminator steps between generator updates is simple. The justification for this approach is empirically justified with some perhaps straightforward experiments and observations. There is no theoretical justification, but this may be expected (albeit perhaps the challenge associated with such a theoretical justification could have been pointed out).

The results have been shown on MNIST and FMNIST which are pretty small datasets.  I understand training GANs is very computationally expensive, but it might be better if you highlight this limitation in the paper as in other domains analysis on these two datasets would not be perceived as sufficient.

This all said, as a non-expert in GANs, I enjoyed reading the paper and think that it is more than worthy of publication in this workshop.

---

### Meta-Review · Area_Chair_7xMa · 2022-10-19

**Recommendation:** Accept